

# Prioritising non-native fish species for management actions in three Polish rivers using the newly developed tool—dispersal-origin-status-impact scheme

Dagmara Błońska[1,2], Joanna Grabowska[1], Ali S. Tarkan[1,3], Ismael Soto[4] and Phillip J. Haubrock[4,5,6]

[1] Department of Ecology and Vertebrate Zoology, Faculty of Biology and Environmental Protection, University of Lodz, Lodz, Poland
[2] Department of Life and Environmental Sciences, Faculty of Science and Technology, University of Bournemouth, Bournemouth, United Kingdom
[3] Department of Basic Sciences, Faculty of Fisheries, Muğla Sıtkı Koçman University, Muğla, Türkiye
[4] Faculty of Fisheries and Protection of Waters, South Bohemian Research Centre of Aquaculture and Biodiversity of Hydrocenoses, University of South Bohemia in České Budějovice, Vodňany, Czech Republic
[5] Department of River Ecology and Conservation, Senckenberg Research Institute and Natural History Museum Frankfurt, Gelnhause, Germany
[6] CAMB, Center for Applied Mathematics and Bioinformatics, Gulf University for Science and Technology, Hawally, Kuwait

Corresponding author
Dagmara Błońska,
dagmara.blonska@biol.uni.lodz.pl

## ABSTRACT

**Background.** Biological invasions are a major threat to global biodiversity, with freshwater ecosystems being among the most susceptible to the successful establishment of non-native species and their respective potential impacts. In Poland, the introduction and spreading of non-native fish has led to biodiversity loss and ecosystem homogenisation.

**Methods.** Our study applies the Dispersal-Origin-Status-Impact (DOSI) assessment scheme, which is a population-level specific assessment that integrates multiple factors, including dispersal mechanisms, origin, status, and impacts, providing a nuanced framework for assessing invasion risks at local and regional levels. We used this tool to evaluate the risks associated with non-native fish species across three major Polish rivers (Pilica, Bzura, and Skrwa Prawa) and to prioritise them for management actions.

**Results.** Using DOSI, we assessed eight non-native species identified in the three studied rivers: seven in both Pilica and Bzura and four in Skrwa Prawa. The DOSI assessment scheme identified high variability in the ecological impacts and management priorities among the identified non-native species. Notably, species such as the Ponto-Caspian gobies exhibited higher risk levels due to their rapid spread and considerable ecological effects, contrasting with other species that demonstrated lower impact levels and, hence, received a lower priority for intervention.

**Conclusion.** The adoption of the DOSI scheme in three major rivers in Poland has provided valuable insights into the complexities of managing biological invasions, suggesting that localised, detailed assessments are crucial for effective conservation strategies and highlighting the importance of managing non-native populations locally.

## INTRODUCTION

Non-native species actively or passively translocated by human actions in regions they have no evolutionary history with (*Soto et al., 2024*), are recognised among the major threats to global biodiversity, affecting all aspects of ecosystems (*Simberloff et al., 2013*; *Cepic, Bechtold & Wilfing, 2022*). These impacts are modulated and often magnified by synergistic interactions with other drivers such as habitat loss, which is considered 'immense, insidious and usually irreversible' (*Strayer, 2010*; *Caffrey et al., 2014*). Freshwater ecosystems are, among all ecosystems, the most vulnerable to being affected by external drivers such as climate change, pollution, and biological invasions (*Havel et al., 2015*; *Haubrock et al., 2021*; *Cuthbert et al., 2023*). Moreover, in the last three decades, biodiversity declined faster in freshwater ecosystems than in marine and terrestrial ecosystems (*Collier, Probert & Jeffries, 2016*; *Reid et al., 2019*, but see *Van Klink et al., 2020*), with non-native species introductions being among the main extinction drivers (*Blackburn et al., 2014*). The intrinsic connectivity of freshwater ecosystems due to *e.g.*, the canalization of large rivers, facilitates the spread of non-native species and ultimately increases the homogenisation of ecosystems (*Marr et al., 2013*). Consequently, mitigation of the effects of non-native species has become one of the most pressing problems ecologists, decision makers, and stakeholders face (*Simberloff, 2015*).

Considering the growing distribution of countless non-native species and the increasing evidence of their staggering negative effects on recipient ecosystems (*Roy et al., 2023*) that are increasingly difficult to monitor and manage (*Moon, Blackman & Brewer, 2015*; *Crowley, Hinchliffe & McDonald, 2017*), there is a rising need for reliable, accessible, and robust tools to assess the potential threat different populations of these non-native species present. Within the last two decades, several protocols (including both 'risk assessment protocols' (*Hawkins et al., 2015*) and 'risk screening' (a.k.a. 'risk identification') (*Vilizzi et al., 2022*); please also see *Srebaliene et al. (2019)* and *Hill et al. (2020)* for a comparison of impact and risk assessment methods) have been developed and implemented worldwide, targeting various taxonomic groups and evaluating current and potential impacts of non-native species. Most of the available assessment protocols share a common feature: they enable the classification of non-native species based on the level of risk they do or may present to a specific assessment area. They, however, differ in complexity (*e.g.*, number of assessed aspects of the species), the underlying scoring system, and the range of impacts assessed. However, although they are designed and tested by scientific experts, a recently conducted—yet critized—comprehensive consistency analysis revealed considerable inconsistency among taxonomic groups, scoring systems, expertise of assessors, and impact evaluated (environmental only or with socio-economic; *González-Moreno et al., 2019*). One pressing issue is that most of these protocols are employed at the national (*Tarkan et al., 2017*) or continental scale (*Haubrock et al., 2021*; *Vilizzi et al., 2021*), which is valuable for national information systems or larger political entities like the European Union but lacks granularity considering the variability of non-native species populations (*Haubrock et al., 2024*). These generalised approaches can lead to underestimating or overestimating impacts at particular sites by assuming that local effects can be generalised at the species level and

be superimposed across regions and ecosystems with similar conditions. *Vice versa*, an assessment at the national scale, even when informed by local risk screenings, may still underestimate the threat a non-native species presents at specific sites, as generalisations across regions and ecosystems with similar conditions may overlook critical local variations. Another important issue is that several of the currently available protocols consider only environmental impacts (*González-Moreno et al., 2019*) as socio-economic impacts are usually difficult to quantify due to the lack of information, despite it being widely accepted that the economic consequences of biological invasions prerequisite an efficient allocation of financial resources *e.g.*, management actions (*Lodge et al., 2016*; *Bang et al., 2022*; *Soto et al., 2023*; *Tarkan et al., 2024*). This further underlines the urgency to easily differentiate and prioritise non-native species for management interventions, resulting in more efficient actions (*Lodge et al., 2016*).

In Poland, over 60% (17 out of 28) of non-native freshwater fish species were introduced more than three decades ago and now form self-sustained populations in the wild or do not breed in natural condition but are keep in aquaculture and used for stocking several water bodies (*Grabowska, Kotusz & Witkowski, 2010*). One of the most important pathways aiding the range extension of non-native aquatic species in inland waters of Poland is the European central invasion corridor (*Jazdzewski, 1980*; *Bij de Vaate et al., 2002*; Fig. 1). This route was used by several non-native fish species to spread in Polish inland waters (*Grabowska, Pietraszewski & Ondračková, 2008*; *Semenchenko et al., 2011*). In response to changing temporal invasion dynamics of non-native species in Polish freshwater ecosystems, alongside recent European Union regulations, the national project run by the government institution *The General Directorate for Environmental Protection* was completed in 2018. It aimed to determine the degree of invasiveness of non-native species in Poland and identify species that pose the greatest threat to invaded ecosystems. To achieve that goal the Harmonia$^+$ protocol was implemented in Poland and named Harmonia$^{+PL}$ (*Tokarska-Guzik et al., 2019*). Among the non-native fish species considered in this recent national project (*General Directorate for Environmental Protection, 2024*), species recently established in Poland include four species of Ponto-Caspian gobies (round, monkey, western tubenose and racer goby; *Neogobius melanostomus*, *N. fluviatilis*, *Proterorhinus semilunaris* and *Babka gymnotrachelus*, respectively), the Chinese sleeper *Perccottus glenii*, and the topmouth gudgeon *Pseudorasbora parva*, but also one species present in European inland waters (including Poland) since the end of the 18$^{th}$ Century, namely the brown bullhead *Ameiurus nebulosus*. The last species included was pirapitinga *Piaractus brachypomus* that is very occasionally recorded as single individuals released by aquarists (*Grabowska, Kotusz & Witkowski, 2010*).

All those species analysed *via* Harmonia$^{+PL}$ protocol, despite their wide distribution across the country (except pirapitinga), were categorised as a low priority in the case of gobies and as a medium priority in the case of Chinese sleeper and topmouth gudgeon. This resulted in the removal of all four goby species from the list of harmful non-native species considered a national Polish concern following the implementation of EU regulations (1143/2014). Furthermore, these changes translate directly into the management of gobies. Although it is still forbidden to introduce them or move them within the environment,

 

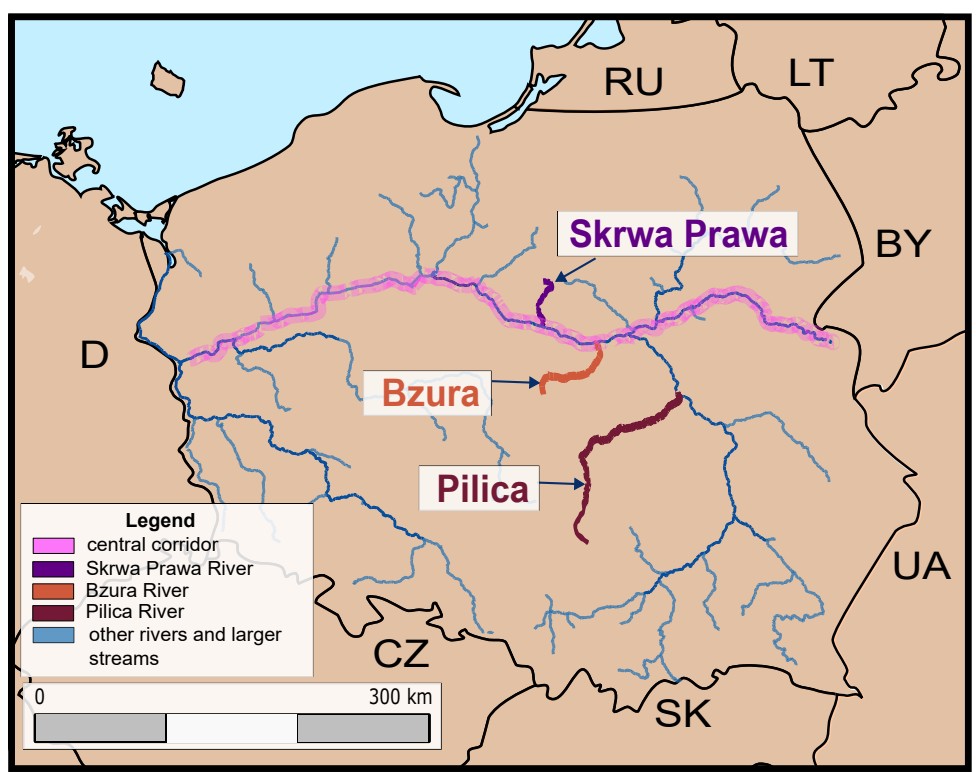

**Figure 1** Map of the rivers (Pilica, Bzura and Skrwa Prawa) assessed using the Dispersal-Origin-Status-Impact (DOSI) scheme.

it is now allowed to keep them (*e.g.*, in the aquarium or private pond), stock, sell, or exchange them. This can, in practice, result in *e.g.*, intentional introductions *via* anglers using gobies as live baits (*Drake & Mandrak, 2014*). Although there is limited evidence of monkey, western tubenose, and racer goby negatively affecting ecosystems they are introduced to *Grabowska et al. (2023)*, this is not the case for the round goby (*Cerwenka et al., 2023*). Thus, the only fish species among the non-native species currently occurring in Polish waters that remained on the lists of Union or Polish concern are the Chinese sleeper, the topmouth gudgeon, the pumpkinseed (*Lepomis gibbosus;* which was not assessed by Harmonia[+PL]), and the brown bullhead (EU regulations 1143/2014 and its implementation at the national level in Dz. U. 2021 poz. 1718).

However, there is growing recognition that biological invasions are context-specific, with considerable variations in the potential of individuals to spread and exert impacts among populations influenced by diverse environmental and biological factors (*Soto et al., 2024*; *Haubrock et al., 2024*). Consequently, there is a need for accurate and standardised assessment protocols that consider the varied effects (both presence and impact) within populations of the same species. The first steps have already been made by *Soto et al. (2024)*, who sorted out the confusion in biological invasion nomenclature and proposed a new assessment scheme—The Dispersal-Origin-Status-Impact (DOSI). The advantage of this approach stems from its thorough yet adaptable framework, which can be applied

to specific populations or at broader regional or ecosystem scales in precise and scientific communication. Therefore, some populations might be identified at different scales of prioritisation and can change over time due to the inherent temporal dynamics of an invasion (*e.g.*, population expanding or higher impacts). DOSI improves upon previous management practices by assisting stakeholders and managers, who often face resource constraints (*Adelino et al., 2021*) in selecting non-native species populations for management actions, thereby enabling them to assess and prioritise non-native species.

To test the relevance and applicability of DOSI, we applied it to non-native species, in three Polish rivers: Bzura, Pilica, and Skrwa Prawa, tributaries of the Vistula River, *i.e.,* the Polish section of the European central corridor of invasions aiming to assess different populations of non-native fish species in rivers of different size. For this, monitoring studies were conducted at least twice on each river, allowing to document several non-native species by examining the entire length of the rivers (*i.e.,* from their sources to their mouths), enabling to obtain an understanding of ongoing changes in the distribution and abundance of these species. The DOSI scheme implementation should provide insight into the threat of non-native species at the population level, enable comparisons with results from the previously conducted Harmonia$^{+PL}$ to identify potential discrepancies and thereby direct future management efforts to particular localities. The DOSI application may also reveal variability in the level of risk that different populations of the same non-native species may pose in different water bodies, as the population level is usually overlooked by more general metrics (*e.g.*, Harmonia$^{+PL}$).

## MATERIALS AND METHODS

### Study sites and data collection

Data for the current study consisted of results published in the national journal issued by the Polish Angling Association (*Scientific Annual of the Polish Angling Association*; *Głowacki et al., 2024*; *Jażdżewski et al., 2012*; *Penczak et al., 2006*) and unpublished data from monitoring the Pilica, Bzura and Skrwa Prawa Rivers (Fig. 1) performed by the Department of Ecology and Vertebrate Zoology, University of Lodz, in 2013 and 2018. They are all tributaries of the Vistula River, however, they differ in length and catchment area (Pilica 332.5 km, 9,258 km$^2$; Bzura 166 km, 7,788 km$^2$; Skrwa Prawa 117.6 km, 1,704 km$^2$). Each of the analysed rivers, the Pilica, Bzura, and Skrwa Prawa, were sampled using the same methodology. One-run electrocatch per constant unit effort (CPUE) was conducted using certified equipment. The effort unit was established following Becklemishev's rule (*Backiel & Penczak, 1989*; *Penczak, 1967*), which asserts that the sampling site length is adequate if no new species are collected with further sampling. Electrofishing was performed by two persons, each using an anode with a dip net from the boat or by wading, depending on the river depth.

The Pilica River was sampled in 2003–2005 (*Penczak et al., 2006*) and again in 2014–2017 (*Głowacki et al., 2024*) at 64 sites along the river; results from previous decades of sampling are also presented in *Penczak et al. (2006)*. Data for the Bzura River were collected in 2013 (unpublished) and 2009–2011 (*Penczak et al., 2012*) from 15 and 17 sites, respectively. The Skrwa Prawa was sampled in 2002–2003 and 2010–2011 (*Jażdżewski et al., 2012*) at 18 sites.

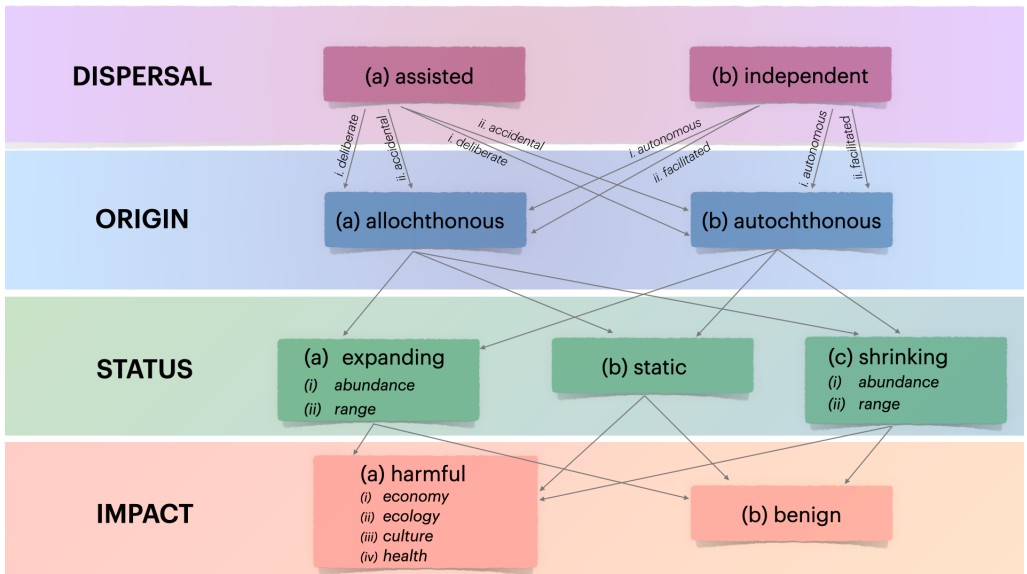

**Figure 2  Flow diagram illustrating the proposed classification scheme for populations entering a novel environment.** A species' **D**ISPERSAL mechanism can be assisted from its place of origin either *deliberately* ($a_i$) or *accidentally* ($a_{ii}$), or it can migrate *independently* of direct human intervention ($b_i$) by being *facilitated* or by exploiting human-driven environmental changes ($b_{ii}$), such as canals. The **O**RIGIN of a species that has its distribution shifted according to the mechanisms described can be *allochthonous* (2a) (not from 'here', with 'here' defined by the spatial scale of interest) or *autochthonous* (2b) (from 'here', as with local species moving within the region of focus). The definition of *allochthonous* or *autochthonous* can also depend on the time elapsed since the species' arrival (*e.g.*, geological time, ancient introductions). **S**TATUS refers to the state of the species' population(s), defined by *abundance* or *range* size (*expanding, static,* or *shrinking*). These assessments depend on the duration of the species' presence, the measurement effort applied to assess population change, and the effectiveness of interventions (if any). The **I**MPACT category assesses whether the species causes harm to one or more sectors (ecology, economy, culture, human health). This assessment ranges from little to extensive harm or determines if the species is benign (no effect).

## The Dispersal-Origin-Status-Impact assessment scheme

The DOSI assessment scheme (Fig. 2) exclusively focuses on negative impacts, emphasizing that these potential threats are significantly more important and distinct than any potential benefits (*Carneiro et al., 2024*). However, DOSI's objective is to prioritise non-native populations for management interventions by considering local risks only, without considering the feasibility or availability of appropriate methods, or the species' potential to spread beyond their current locations. The focus on the population level distinguishes DOSI from other assessment tools, like the Harmonia$^{+PL}$ protocol, that are commonly applied at varying regional scales (*i.e.*, assessment regions) without a strict focus on the population level. The Harmonia$^{+PL}$ protocol looks at non-native species at the national level and consists of 30 questions divided into the two main modules "invasion process" and "impact" and a final score calculated based on combined results obtained for both modules.

DOSI prioritisation is structured around a hierarchy of primary dispersal mechanisms, distinguishing between non-native populations that can (a) spread independently and

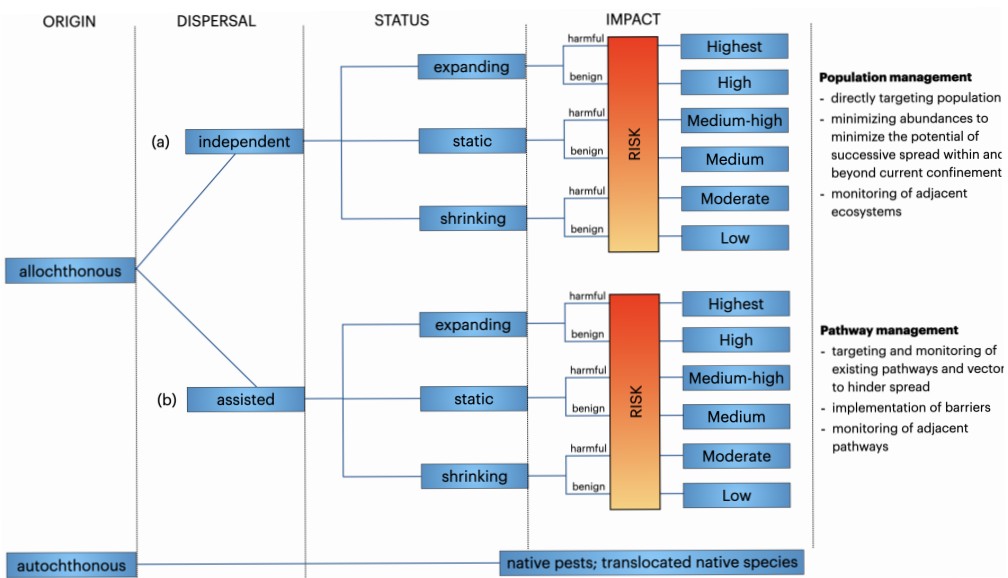

**Figure 3** Priority ranking for management interventions of non-native populations based on the Dispersal-Origin-Status-Impact (DOSI) assessment scheme (**Supplement S2**). (A) Populations dispersing primarily without human assistance, and (B) populations dependent on human assistance for dispersal. See supplement figure for a definition of the various priority classes.

invade areas beyond the introduction site, (b) rely mainly on human assistance and the presence of pathways and vectors, (c) have the capability for both assisted and independent spread (*i.e.,* evaluated for both a and b), and (d) the populations' status, which defines the state of a population within the target site and the local impact it exerts. This means, that populations that can spread independently and with assistance, and those showing changes in abundance and range, are ranked higher than those with only one type of dependency. This is because the former scenarios indicate a greater and more harmful invasion potential. Similarly, populations with one static and one expanding dependency are also ranked higher. Conversely, if a population is determined to have no known local impact, it is lowered in the priority ranking and thus requires a different response (Fig. 3).

To test the DOSI assessment scheme for each river, we considered all non-native fish species identified. We assessed each identified non-native fish species in the Pilica River (seven non-native species), the Bzura River (seven non-native species), and the Skrwa Prawa River (four non-native species; Table 1) using DOSI to provide an objective overview for the prioritisation of each rivers' non-native species populations (Fig. 3). Information on changes in abundance growth or range extension were not always precise based on the field samplings, thus we filled information gaps based on our expert knowledge of the study sites and the respective non-native species invasion histories. Consequently, we discussed the DOSI assessment outcomes for the assessed species with the previous screening based on Harmonia[+PL] to identify discrepancies and ultimately test if the population level considered by DOSI provides relevant variability.

**Table 1  Summary of non-native fish species occurrence found in each river (Pilica, Bzura and Skrwa Prawa).**

| Species | Common name | Pilica | Bzura | Skrwa Prawa |
|---|---|---|---|---|
| *Babka gymotrachelus* | racer goby | + | + | + |
| *Neogobius fluviatilis* | monkey goby | + | + | − |
| *Proterorhinus semilunaris* | western tubenose goby | + | + | + |
| *Percottuss glenii* | Chinese sleeper | + | + | + |
| *Ameiurus nebulosus* | brown bullhead | + | − | − |
| *Carassius gibelio* | gibel carp | + | + | + |
| *Pseudorasbora parva* | topmouth gudgeon | + | + | − |
| *Cyprinus carpio* | common carp | − | + | − |

## RESULTS

Within the three tested rivers (*i.e.,* Pilica, Bzura and Skrwa Prawa rivers), eight non-native fish species were identified, three of which (*i.e.,* the monkey goby, racer goby, and western tubenose goby) were of Ponto-Caspian origin, another three (*i.e.,* the topmouth gudgeon, Chinese sleeper, and gibel carp) originated from Eastern Asia, while one (brown bullhead) originated from North America, and another one (common carp) from the Danube catchment (Tables S1–S3). All goby species as well as Chinese sleeper and topmouth gudgeon in each evaluated river (Pilica, Bzura, Skrwa Prawa) were classified as independently dispersing, whereas brown bullhead, gibel carp, and common carp as spreading depending on human assistance.

The DOSI ranking was not consistent among species and rivers highlighting the context-dependency of invasions (Fig. 4). The monkey goby was designated as Highest Priority in the Pilica and Bzura Rivers (the species was absent in the Skrwa Prawa and could not be evaluated there) due to its increasing range and abundance leading to competitive pressure on native species (*Błońska et al., 2016*; Błońska et al. under review). The monkey goby ranking was constant across the analysed sites (Highest). The second one was the western tubenose goby, which also received the status High Priority in all rivers. Although the species is also continually extending its range and abundance, no negative impact has been observed (yet). The third was the topmouth gudgeon, which was ranked as Medium Priority based on static range and abundance in both Pilica and Bzura.

Both the racer goby and the gibel carp were ranked as Highest Priority in the Skrwa Prawa and Pilica River, respectively. In other sites, both species were ranked as High or Medium Priority. These discrepancies result from the inconsistent dynamics of both species. Besides range extension and abundance increase, they displayed a negative effect on native biota at one site while having no influence at another (*e.g.,* gibel carp in the Pilica *vs.* Bzura River). The Chinese sleeper was scored as Medium Priority in both the Pilica and Bzura Rivers and High Priority in the Skrwa Prawa, where its abundance was increasing rather than static. The only species designated with Low Priority was the brown bullhead, whose decreasing range and abundance are probably due to the less suitable riverine habitat for this species compared to more stagnant waters such as oxbow lakes.

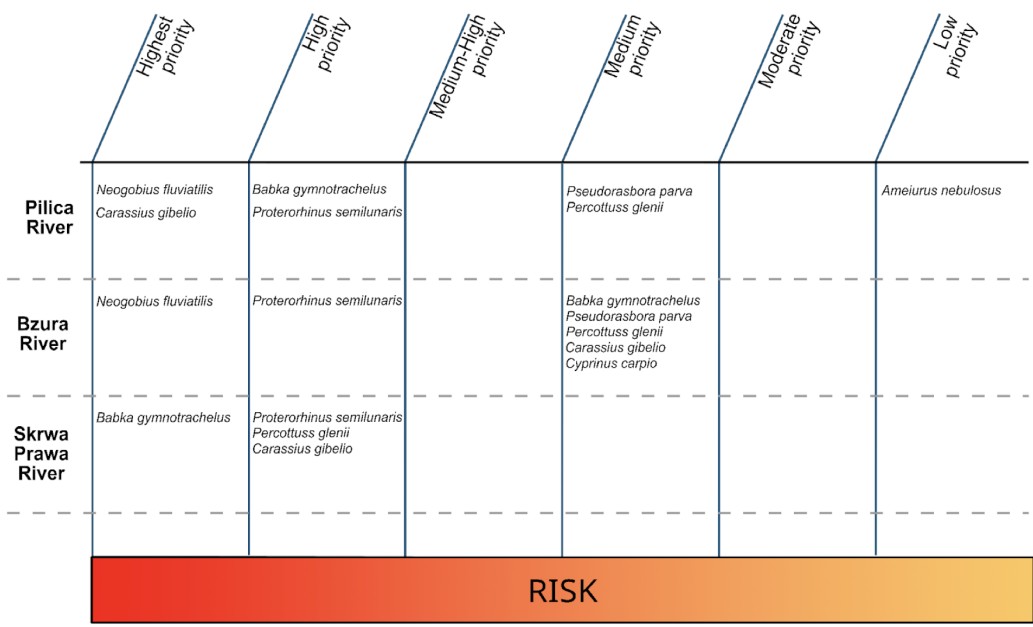

**Figure 4** Ranking of established non-native fish species for management targeting populations in (a) Pilica, (b) Bzura and (c) Skrwa Prawa Rivers following the assessment with the Dispersal-Origin-Status-Impact (DOSI) scheme.

**Table 2** Comparison of Dispersal-Origin-Status-Impact (DOSI) assessment scheme ranking and applied in Poland in 2018 Harmonia[+PL] assessment of non-native freshwater fish in three evaluated rivers Pilica, Bzura and Skrwa Prawa.

| species | DOSI (Pilica) | DOSI (Bzura) | DOSI (Skrwa Prawa) | Harmonia[+PL] |
|---|---|---|---|---|
| *Babka gymotrachelus* | High | Medium | Highest | Potentially invasive |
| *Neogobius fluviatilis* | Highest | Highest | — | Potentially invasive |
| *Proterorhinus semilunaris* | High | High | High | Potentially invasive |
| *Percottuss glenii* | Medium | Medium | High | Moderately invasive |
| *Ameiurus nebulosus* | Low | — | — | Moderately invasive |
| *Carassius gibelio* | Highest | Medium | High | — |
| *Pseudorasbora parva* | Medium | Medium | — | Moderately invasive |
| *Cyprinus carpio* | — | Medium | — | — |

The DOSI ranking differentiated among populations (ranging from High to Low Priority) and was not complementary with the Harmonia[+PL] results, which differentiated the six previously assessed species into moderately invasive and potentially invasive (Table 2).

## DISCUSSION

In the current study, we evaluated the risk posed by non-native species in three temperate lowland rivers in Poland (the Pilica, Bzura and Skrwa Prawa River) by applying the DOSI

scheme (as assessment protocol) and comparing our results to the Harmonia$^{+PL}$ screening protocol outcomes. Although DOSI and Harmonia$^{+PL}$ are not directly comparable for this reason, as DOSI focuses on population-level prioritization while Harmonia$^{+PL}$ identify species risk based on broader ecological implications of species introduction, this distinction allows DOSI to provide more granular, site-specific risk rankings that may vary across locations. This variability was reflected in our findings, where species were not consistently assigned the same rank across the three rivers, demonstrating how DOSI can capture localized differences in species impact, even within similar ecological contexts. Across the three rivers, species were not always designated with the same rank. High and Medium Priority ranks dominated (six times each) with four Highest Priority and only one Low Priority, underlining the DOSI's ability to prioritise non-native species at the population level.

## Population-level assessment

Among the non-native species surveyed using DOSI were fish strongly associated with riverine habitats, specifically Ponto-Caspian gobies. The evaluated rivers are in proximity to the Central invasion corridor in Europe, which serves as the main expansion route for these species in Poland (*Semenchenko et al., 2011*). Both monkey and western tubenose gobies were designated as Highest and High Priority, respectively, constantly occurring across considered rivers that resulted from increasing range and abundance. They are among the fastest spreading non-native species in Poland, with the monkey goby having extended its range by 340 km in the last five years (*Bylak & Kukuła, 2024*) and the tubenose goby by 255 km in seven years (*Grabowska et al., 2021*). Once established, they often become abundant and may pose a threat to native species due to competition (*Borcherding, Heubel & Storm, 2019*; *Błażejewski et al., 2022*) and predation (*Grabowska et al., 2023*), even though they do not display aggressive behaviour (*Van Kessel et al., 2011*; *Błońska et al., 2016*; *Błońska, Kobak & Grabowska, 2017*). Although they do not affect native species directly, their high abundance and similar resource requirements can threaten native species (*Błońska et al., 2016*; Błońska et al. under review). A distinct example is the racer goby, which was ranked differently in each evaluated river, from Medium in Bzura, High in Pilica to Highest Priority in Skrwa Prawa. Although this variability in DOSI rankings among these goby species can likely be explained by differences in habitat requirements (*Plachocki et al., 2020*; *Bylak & Kukuła, 2024*), it should be noted that racer goby is not as efficient in expanding its range as monkey and tubenose gobies, but it can significantly affect recipient communities (*Grabowska et al., 2023*). Observations under laboratory conditions for instance revealed that racer gobies aggressively outcompete native species when resources are limited (*Kakareko et al., 2013*; *Grabowska et al., 2016*). This adverse effect on native species was also observed in the field (*Kakareko et al., 2016*). Impact of racer goby was not confirmed directly in the analysed rivers, however, its extending range and abundance ranked it with higher priority in Pilica and Skrwa Prawa, which in the case of the latter one was reflected by decrease in population of white-finned gudgeon (*Romanogobio albipinnatus*), golden loach (*Sabanejewia baltica*) and European bullhead (*Cottus gobio*).

Another group of assessed non-native species consisted of species that naturally express a preference for stagnant waters and often occur in various natural and artificial water bodies in the vicinity of river valleys from where individuals or in relatively small groups may accidently enter a main course of a river. Some of them, like the Chinese sleeper, are locally very common and even dominate in some oxbow lakes or other parts of flood plains (*Koščo et al., 2003*; *Grabowska et al., 2011*; *Reshetnikov, 2013*; *Rechulicz, Płaska & Nawrot, 2015*) where water current is slower or even blocked like in old side arms, bays or marinas, and occasionally are flushed to the main river channel during high water episodes. It is claimed that the Chinese sleeper uses rivers for fast long-distance dispersal during floods (*Reshetnikov, 2013*). It also occurs as an accidentally introduced species in fish ponds and spreads with stocking material of commercial species (*Reshetnikov, 2013*; *Grabowska et al., 2020*). We acknowledge that the frequency of this species' reporting in rivers, but also that of numerous other non-native fish species, will increase in the foreseeable future (*Witkowski & Grabowska, 2012*; *Seebens et al., 2021*). However, the opposite may be the case for the brown bullhead that used to occupy similar types of waters as the Chinese sleeper but its range and abundance have decreased in Poland since the 1980s, when its intentional introductions by local angling associations were very common (*Witkowski, 1996*; *Grabowska, Kotusz & Witkowski, 2010*) but nowadays is treated as a "pest" to be removed (Harmonia[+PL]; *Grabowska, Kakareko & Mazurska, 2018c*). This ultimately underlines the importance of local assessment for non-native species.

The assessed non-native species also include species that, in most cases, directly originated from fish ponds and accidentally escaped to adjacent streams and rivers. One of them is the gibel carp, a cosmopolitan, eurytopic species; currently being the most widespread non-native fish in Poland's inland waters (*Witkowski, 1996*; *Grabowska, Kotusz & Witkowski, 2010*). In fish ponds, it is often stocked with accompanying carp, and it is introduced into special types of commercial fishery, *i.e.,* "put-and-take" recreational angling ponds (*Grabowska, Kotusz & Witkowski, 2010*). Another non-native species found in fish ponds, the topmouth gudgeon, spreads unintentionally in its non-native range as a contamination of stocking material of other Asian cyprinids, such as common carp and silver carp (Hypophthalmichthys molitrix) (*Witkowski, 1996*; *Grabowska, Kotusz & Witkowski, 2010*; *Gozlan et al., 2010*). Both gibel carp and topmouth gudgeon are often found in rivers in a large abundance, particularly after cleaning and other maintenance practices in fish ponds (*Witkowski, 2009*; *Takács et al., 2017*). However, such a situation was not observed in the studied rivers as only single or few individuals of these species were caught during the sampling.

Although there is some evidence that species like Chinese sleeper, gibel carp, and topmouth gudgeon have impacts on native species, economy, and even culture in stagnant waters (*Gozlan et al., 2010*; *Tarkan et al., 2012*; *Kutsokon et al., 2021*), their ephemeral presence in rivers do not create a serious threat for riverine ecosystems. Thus, they were scored as low or medium priority due to a lack of abundance growth and impacts. However, these species are currently expanding their invasive ranges and must be treated with consciousness and their occurrence in rivers should be monitored.

## DOSI and Harmonia[+PL]

The impact of non-native species can differ substantially across sites, generalising at larger geographically or political scales complicated or even flawed (*Haubrock et al., 2024*). Here, we found substantial differences in the scores non-native species obtained across the three studied rivers, and, considering the number of Highest and High Priority species, DOSI even suggested that the Pilica and Skrwa Prawa Rivers are under higher pressure than the Bzura River, where most species was identified as of Medium Priority (5 out of 7). DOSI also identified noteworthy differences to Harmonia[+PL], which is applied at the country level and previously assessed all non-native species that were also assessed by DOSI in this study (except for the gibel and common carp). Indeed, the highest discrepancies were among Ponto-Caspian gobies, which were assigned a High or Highest Priority in most analysed rivers following DOSI, while in Harmonia[+PL] they were ranked as potentially invasive non-native species (*Grabowska, Kakareko & Mazurska, 2018b*; *Kakareko, Grabowska & Mazurska, 2018a*; *Kakareko, Grabowska & Mazurska, 2018b*). It can be partly explained by the differences in scoring scheme applied in DOSI and Harmonia[+PL] assessment.

Thus, even that in the case of Ponto-Caspian gobies they got the highest score assessing their invasion process (what indicated that at time of the assessment they were still in the expansion phase with a high risk of further spread), their "impact" was scored as low or moderate and it influenced the final risk assessment score. It resulted from the fact that the knowledge of the impact of that species on biota and inanimate elements of the ecosystem was low or there were not convincing studies proving such potential impact (reviewed in *Grabowska et al., 2023*). A contrasting case was the brown bullhead, recorded only in one of three analysed rivers and accordingly only ranked as Low Priority by DOSI, was assessed as a moderately invasive non-native species in Harmonia[+PL] (*Grabowska, Kakareko & Mazurska, 2018c*). Topmouth gudgeon and Chinese sleeper received similar scores (moderate) in both protocols (*Grabowska, Kakareko & Mazurska, 2018a*; *Kakareko, Grabowska & Mazurska, 2018c*). Those three species got much higher scores in the "impact" module of Harmonia protocol which increased the results of their risk screening.

## Management following DOSI

The findings from the DOSI scheme highlight the importance of distinguishing between non-native species that spread independently and those that spread through human assistance. This differentiation is crucial for developing effective management strategies tailored to the specific mechanisms of spread for each species. In the evaluated rivers, five species have been identified to spread independently, whereas three species have been spreading primarily through human assistance.

For species that spread independently, such as Ponto-Caspian gobies and Chinese sleeper in River Pilica, Bzura and Skrwa Prawa, population management is essential. Effective strategies should focus on the decimation of the population by implementing targeted removal programs to reduce the population size, limiting propagule and colonization pressure through measures such as ecosystem restorations to make the environment less conducive for these species to reproduce and spread (*Dorenbosch et al., 2017*), and lowering exerted impacts by ongoing monitoring and intervention to mitigate the negative

impacts on native species and ecosystems. Current efforts in some regions, such as existing management actions, have already shown success in lowering the abundances of these species (*e.g.*, *Dorenbosch et al., 2017*). Continued and enhanced efforts are necessary to ensure long-term control and protection of native biodiversity (*Leuven et al., 2017*).

For species spreading through human assistance, such as gibel carp in River Pilica, managing the pathways of introduction is critical. Relevant pathways include monitoring and regulating the transport and release of fish stock to prevent contamination with non-native species, educating and regulating activities such as fishing and boating to reduce unintentional introductions, and ensuring that water management practices, such as the maintenance of fish ponds and river channels, do not inadvertently facilitate the spread of non-native species. Effective management of these pathways is possible through stringent regulation, public education, and collaboration between stakeholders, including local communities, conservation organizations, and government agencies.

Based on the DOSI assessment, it is recommended to enhance monitoring and research, as continuous monitoring and research are essential to track the spread and impact of non-native species. Implementing targeted management plans for high-priority species in each river is also crucial. Increasing public awareness and involvement through education and engagement in monitoring and control activities is necessary, as well as strengthening regulations and enforcement to prevent the introduction and spread of non-native species through human activities. By addressing both independent and assisted spread, we can develop a comprehensive approach to managing non-native species and protecting the integrity of river ecosystems in Poland.

## CONCLUSION

The application of the DOSI scheme in evaluating the risk posed by non-native species in three temperate lowland rivers in Poland demonstrates that ranking non-native species is both feasible and effective. The study highlights substantial differences between DOSI's population-level assessments, and the species-level risk screening provided by Harmonia$^{+PL}$. These differences underscore the importance of localized and population-specific evaluations in understanding and managing non-native species. DOSI's ability to assess the risk at the population level provides nuanced insights that are critical for effective management. By identifying the specific threats and prioritising non-native species based on their local impact and spread, DOSI enables more targeted and relevant management decisions. This approach helps in determining the most appropriate management strategies, whether it involves population management for independently spreading species or pathway management for those spreading through human assistance. Applying DOSI in combination with monitoring surveys could enhance the accuracy and timeliness of risk assessments, allowing for more proactive intervention strategies. Expanding the application of DOSI to other geographical regions and aquatic environments may reveal further insights into its effectiveness in addressing varying ecological contexts.

# ACKNOWLEDGEMENTS

The authors thank Bartosz Janic for his support in preparing the map for Fig. 1. The authors would like to thank Lidia Marszał, Tadeusz Penczak, Mirosław Przybylski, Mariusz Tszydel, Grzegorz Zięba, Dariusz Pietraszewski, Bartosz Janic, Szymon Tybulczuk, Maciej Jażdżewski, Łukasz Kapusta, and Andrzej Kruk for their assistance in the sampling in 2013 and 2018 on the Bzura River.

## Funding
The authors received no funding for this work.

## Competing Interests
The authors declare there are no competing interests.

## Author Contributions
- Dagmara Błońska conceived and designed the experiments, performed the experiments, analyzed the data, prepared figures and/or tables, authored or reviewed drafts of the article, and approved the final draft.
- Joanna Grabowska conceived and designed the experiments, performed the experiments, analyzed the data, prepared figures and/or tables, authored or reviewed drafts of the article, and approved the final draft.
- Ali S. Tarkan conceived and designed the experiments, performed the experiments, analyzed the data, prepared figures and/or tables, authored or reviewed drafts of the article, and approved the final draft.
- Ismael Soto conceived and designed the experiments, performed the experiments, analyzed the data, prepared figures and/or tables, authored or reviewed drafts of the article, and approved the final draft.
- Phillip J. Haubrock conceived and designed the experiments, performed the experiments, analyzed the data, prepared figures and/or tables, authored or reviewed drafts of the article, and approved the final draft.

## Data Availability
The data is available in the Supplementary File.

## Supplemental Information
Supplemental information for this article can be found online at http://dx.doi.org/10.7717/peerj.18300#supplemental-information.

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
