# Peer review of "Prioritising non-native fish species for management actions in three Polish rivers using the newly developed tool—dispersal-origin-status-impact scheme"

_PeerJ, doi:10.7717/peerj.18300_

## Round 0.1 · original submission · Major Revisions

First of all my apologies for the long review process. We had a rather difficult time finding reviewers. I have now received two reviews and encourage you to address all their comments. Please note that one of them attached an annotated manuscript file with more detailed comments. I agree with the reviewer assessments but reviewer 1 makes some more fundamental remarks that could be addressed in two ways: makes the changes suggested or provide a comprehensive response as to why not.

·

Basic reporting

Please refer to the annotated PDF that I have attached.

The core issue with this MS is indicated in my comment at lines 213-215, which I paste in here below:

There is therefore a fundamental flaw at the core of this MS: The 'comparison' between the Harmonia+PL outcomes for the species under study with the results of the DOSI scheme. This is in fact an 'apples to oranges' comparison.
Given the major issue above, the entire MS must be re-organized to eliminate the flawed 'one-to-one' comparison between the two different kinds of protocols and replace it with a discussion on how the outcomes of Harmonia+PL have affected the results of the DOSI protocol.
Overall, this will not involve any major re-write of the MS, but just an 'adjustment' to reflect the sequence between risk screening and risk assessment as part of the risk analysis process.
The Authors can refer to Vilizzi et al. (2022):
Vilizzi L., Hill J.E., Piria M., Copp G.H. 2022. A protocol for screening potentially invasive non-native species using Weed Risk Assessment-type decision-support tools. Science of the Total Environment 832: article 154966. https://doi.org/10.1016/j.scitotenv.2022.154966
for an explanation of the sequential step in the non-native species risk analysis process and provide a short description of it in the paragraph at lines 185-194.

Experimental design

Sampling has been carried out correctly.

Validity of the findings

High validity.

Additional comments

Please refer to annotated PDF for all comments.

Reviewer 2 ·

Basic reporting

The ms “Prioritising non-native fish species for management actions in three Polish rivers using the newly developed tool – Dispersal-Origin-Status-Impact DOSI scheme” by Blonska et al. examines the suitability of a recently developed index for characterizing the invasiveness and potential risk of certain non-native fish species in three rivers. Additionally, authors compared the results of this index with those of another index previously developed for larger scale assessments. The ms is well-written with proper English.

My concerns with the ms are as follows:

1. The second dataset for this study were collected 7-13 years ago. DOSI values were calculated on this dataset, but the Discussion section presents these results as recent values. Do authors not think that the studied species’ expansion range, impacts etc have changed since then especially in light of that authors indicated that for example monkey goby extended its range more than 300 km within 5 years?

2. I feel that sorting the studied species into groups (e.g. independently dispersing vs. human assistance) is rather subjective. For example, gibel carp is well-known to spread without human assistance. Maybe the species was introduced for commercial reasons, but after its establishment, it can easily spread without any human-mediated actions. More information on the species grouping would be needed (e.g. more references)

3. Description of the scoring system of DOSI is a bit unclear in the ms. Monkey goby, for example, got higher values along the invasiveness gradient, while Chinese sleeper has lower values. Authors indicated that the reason behind this pattern is that monkey goby spread faster. However, authors also indicated that “impact” is also very important when calculating ranks. Chinese sleeper, while does not spread rapidly, has been shown to negatively impact native species while we have less information about monkey goby’s influence. I think more information would be needed for readers in the text to understand how important the background variables were in rating.

4. The Conclusion section contains mainly repetitions of the Discussion. I think it can be deleted.

Minor suggestion:

line 56: authors indicated that non-native species can spread without human assistance through the whole ms. Revise this sentence.
line 104: Fig 1 can be added here.
line 116: use Chinese sleeper, since “Amur sleeper” occurs only here
line 124: pumpkinseed was not indicated above
line 192: start the sentence with Harmonia protocol instead of “this” to clear which scoring system you mean.
line 229: authors suggested no well-known impacts of monkey goby earlier. Here they indicated that it can potentially compete with native fishes. This is confusing.
line 238-239: How can we know that these species had a negative impacts on native species at one site but no impacts at an other one?
line 266: and maybe predation on fish eggs?
line 357: what types of habitat modifications would be good?

Experimental design

no comment

Validity of the findings

no comment

Additional comments

no comment

---

## Round 0.2 · accepted · Accept

I believe that all reviewer comments have been sufficiently addressed.